# Changes in the prevalence of diabetes, prediabetes and associated risk factors in rural Baluchistan; a secondary analysis from repeated surveys (2002–2017)

Khalid Abdul Basit[1,2]⊙, Asher Fawwad[2,3]⊙, Nida Mustafa[2‡], Thomas Davey[1]⊙, Bilal Tahir[2‡], Abdul Basit ᴵᴰ[4]*

1 Acute Medicine and Ambulatory Care, Whipps Cross University Hospital, Barts Health NHS Trust, London, United Kingdom, 2 Research Department, Baqai Institute of Diabetology and Endocrinology, Baqai Medical University, Karachi, Sindh, Pakistan, 3 Department of Biochemistry, Baqai Medical University, Karachi, Sindh, Pakistan, 4 Department of Medicine, Baqai Institute of Diabetology and Endocrinology, Baqai Medical University, Karachi, Sindh, Pakistan

⊙ These authors contributed equally to this work.
‡ NM and BT also contributed equally to this work.
* abdulbasit@bide.edu.pk, research@bide.edu.pk

**Data Availability Statement:** All relevant data are within the paper.

## Abstract

To observe trends of diabetes and its associated risk factors from health surveys 2002–2017 in rural areas of Baluchistan-Pakistan and a secondary analysis based on community based health surveys of Baluchistan conducted between 2001–02, 2009–10, and 2016–17. A total of 4250 participants were included in this combined analysis, 2515 from 2001–2002, 1377 from 2009–2010 and 358 from 2016–2017 survey year. In each survey, detailed information of baseline parameters were noted on a predesigned questionnaire. Fasting plasma glucose (FPG) was used for diagnosis of diabetes for comparative purposes in this analysis. Cardiovascular (CVD) risk factors including hypertension, obesity, dyslipidaemia, tobacco use, alcohol consumption, and physical activity were compared. Most subjects were aged 30–50 years and males were found higher in 2016–17 compared to 2001–02 and 2009–10. Pronounced increases in BMI, waist circumference, blood pressure, and family history of diabetes were observed in 2016–17. Diabetes prevalence was 4.2 (3.4–4.9), 7.8 (6.6–9.2) and 31.9 (26.9–37.4), whilst pre-diabetes was 1.7 (1.3–2.2), 3.6 (2.8–4.6) and 10.7 (7.6–14.9) in years 2001–02, 2009–10, and 2016–17, respectively. Among those aged 20-39years, prevalence of diabetes was stable from 2001–10 yet increased considerably between the ages of 30-39years in 2016–17. Throughout the observed period, rapid increases were observed in hypertension, obesity, and dyslipidaemia, however, addiction to tobacco use and alcohol intake decreased. Adjusted odd ratios showed age, marital status, education, hypertension, and family history of diabetes as associated risk factors for glycaemic dysregulation. The rural Baluchistan population is confronted with increasing trends of early onset diabetes due to highly associated CVD risk factors, especially central obesity and dyslipidaemia, raising a major public health challenge.

**Funding:** The author(s) received no specific funding for this work.

**Competing interests:** The authors have declared that no competing interests exist.

## Introduction

Diabetes Mellitus is a heterogeneous disease characterised by impaired glucose regulation or as a consequence of chronic hyperglycaemia [1]. Epidemiological studies in Pakistan from the 1990s have reflected the rising trends of diabetes and prediabetes with a change in the pattern of associated risk factors over this period [2, 3]. In 2021, the latest estimates from the International Diabetes Federation (IDF) confirmed Pakistan is home to around 33 million people with diabetes (aged 20–79 years) and now ranks third for diabetes globally after China and India [4].

Dysglycaemia was remarked as having the highest annual rate of change in disability-adjusted life years (DALYs) in the past decade [5]. People with untreated or poorly controlled diabetes in Pakistan have shown serious micro and macro-vascular complications, and ultimately premature mortality [6]. Pakistan has a vast rural population with diverse economic, cultural, social and educational patterns [7]. Baluchistan is the largest province in the area and sparsely populated in contrast to other provinces of Pakistan. In Baluchistan, around 71% of the population are poor, and 45% are illiterate [8]. The absence of formal education, large family sizes, early and late weaning, avoidance of breastfeeding, poverty, and undernutrition/malnutrition are significant characteristics of people living in Baluchistan [9]. With rising trends of diabetes in many provinces of Pakistan, a recent study from Baluchistan has revealed the appalling frequencies of comorbidities including cardiovascular disease (30%), neuropathy (17.8%), ocular conditions (14.8%), nephropathy (10.7%), cerebrovascular disease (6.8%), and foot conditions (1.5%) [10].

This is mainly due to the increasing burden of hyperglycaemia, hypertension, smoking, unhealthy lifestyles, sedentary behaviour, obesity, and the population's genetic and epigenetic background [10].

The Global Monitoring Framework (GMF) for noncommunicable diseases (NCDs) was developed by the World Health Organization (WHO) in 2013 [11] for the prevention and control of major behavioural and biological risk factors that mainly include reducing hypertension and halting the rise of diabetes and obesity [12]. As a result of the current alarming situation, the accomplishment and monitoring of these targets necessitate robust surveillance [13]. Physical activity, reduction in weight, and pharmacological interventions along with early diagnosis can control not only blood glucose but also complications associated with diabetes, especially in high-risk populations [14]. Therefore, the aim of this study is to compare the data from three health surveys conducted in 2001–02, 2009–10 and 2016–17, and to examine the trends for the prevalence of diabetes, pre-diabetes, and its associated selected cardiovascular risk factors among adults in rural areas of Baluchistan.

## Method

### Study design

These secondary analyses of existing data compared the three community based health surveys conducted in the years 2001–02, 2009–10, and 2016–17 [15–17]. Two health surveys were organised at the same location in sixteen southern Baluchistan villages within a distance of 2.5 km from the city of Hub. The third (2016–17) survey was a nationally representative study, conducted in urban and rural regions of all four provinces of Pakistan [15]. Data from rural Baluchistan from 2016–17 was included in order to compare the same population.

### Inclusion criteria

Permanent, non-pregnant adult residents aged 20 years or above were included in the analysis.

## Study participants

A total of 4250 participants were included in this combined analysis, 2515 from 2001–2002, 1377 from 2009–2010 and 358 from 2016–2017 survey year.

## Survey team

A well-trained team including doctors, health visitors, lab technicians, and volunteers were involved in the surveys. For each study, the team visited the selected household members, explained the detail of the survey, and requested the participants visit the camp on a specified day with overnight fasting of at least 8 to 12 hours [15].

## Diagnostic criteria

In the 2001–02 and 2009–10 surveys, fasting plasma glucose (FPG) was used as a diagnostic criterion for diagnosis of diabetes; in 2016–17, oral glucose tolerance tests (OGTT) were performed (each participant was given a 75-gram anhydrous glucose load) instead. For comparative purposes, the results of FPG from all three survey years were used in the analysis. The American Diabetes Association (ADA) & World Health Organization (WHO) suggest using FPG as diabetes diagnostic criteria for epidemiological studies [15].

## Patient and public involvement

After an informed written consent, subjects were registered and detailed information of their baseline demographic and anthropometric measurements were noted on a predesigned questionnaire. Blood samples were obtained for biochemical parameters that include FBS and lipid profile. In survey year 2001–02, waist circumference and lipid profiles are missing and are not compared.

## Demographic and anthropometric details

Demographic details included marital status (single, married, divorced/separated/widowed), education (illiterate/literate, primary/secondary/high school and above), exercise (sedentary to light, moderate to heavy), family history of diabetes, tobacco use (current addiction and ex-addict) and alcohol addiction (current addict and ex-addict). Using the standardised technique, anthropometry indices such as blood pressure, body weight, height, and waist circumference were measured for all eligible subjects. The weight of subjects was measured in light clothing and without shoes using a digital scale positioned on a flat surface to the nearest 0.1 kg. Height to the nearest 0.1 cm was recorded using a measuring scale in erect posture vertically touching the occiput, heels, back, and hip on the wall. Body mass index (BMI) was calculated by dividing weight in kg with height in $m^2$. Waist circumference (WC) was measured between the centre point of the lower margin of the ribs. Blood pressure was measured in a sitting position after 10 minutes of rest using a mercury sphygmomanometer. In the survey year 2002, details of marital status, education, exercise, alcohol addiction, WC, and lipid profile were missing [17].

## Diabetes and intermediate hyperglycaemia

As per the WHO Criteria: subjects with a FPG $\geq$ 126 mg/dl were recorded as diabetic, FPG between 110–125 mg/dl as having intermediate hyperglycaemia, and a FPG < 110 mg/dl as without diabetes (survey years 2001–02 and 2016–17). As per ADA criteria, FPG between 100 and 125 mg/ dl was taken as intermediate hyperglycaemia and <100 mg/dl as without diabetes

(survey year 2009–10) [15–17]. All the surveys defined known diabetes as previously diagnosed diabetes by a physician and/or taking anti-diabetic agents.

## CVD risk factors

Newly diagnosed hypertension was defined as a systolic or diastolic blood pressure ≥ 130/85 (2009–10) ≥ 140/90 mmHg (2001–02 and 2016–17) [16–18]. Self-reported hypertension was defined as physician diagnosed or using antihypertensive medication. BMI $<23$ kg/m$^2$ was considered as normal, ≥23–24.9 kg/m$^2$ as overweight and ≥25 kg/m$^2$ as obese as per Asia Pacific Guideline [19, 20]. For the current analysis, obesity was categorised as generalised overweight/obesity, central obesity, and combined obesity. Generalised overweight/obesity was defined as a BMI of 23kg/m$^2$ or higher for both males and females. Central obesity was defined as waist circumference $>90$ cm (in males) and $>80$ cm (in females). Combined obesity was considered if the person has generalised obesity and/or central obesity [19]. Dyslipidaemia was classified as having a serum total cholesterol (TC) $>200$ mg/dl, serum low lipoprotein density-cholesterol (LDL-C) $>130$ mg/dl, serum high lipoprotein density-cholesterol (HDL-C) $<40$ mg/dl (for males) and $<50$ mg/dl (for females), and serum triglycerides $>150$ mg/dl or TC to HDL ratio (TC/HDL) as ≥5.9 [21, 22]. Known dyslipidaemia was defined in subjects already using medications to control dyslipidaemia. Current tobacco was defined as the use of tobacco irrespective of duration and quantity consumed and ex-tobacco users as having used tobacco in the past (but not in the preceding month), and those who never used any kind of tobacco were defined as non-tobacco users [23]. No physical activity or only once a week was defined as sedentary to light, and two to three times per week or greater as moderate to heavy [19].

## Ethical approval and informed consent

Ethical approval for survey years 2001–02 and 2009–10 was obtained from the Institutional Review Board (IRB) of Baqai Institute of Diabetology and Endocrinology (BIDE), Baqai Medical University (BMU) [16, 17]. For 2016–17, ethical approval was obtained from the National Bioethics Committee (NBC) of Pakistan (Ref: No.4-87/17/NBC-226/NBC/2664) [15]. In all survey years people were informed in advance, and written informed consent was taken before collecting the blood samples.

## Statistical analysis

Estimates of prevalence of cardiovascular risk factors were age standardised by using the 1998 census for the respective population available at the Pakistan Bureau of Statistics [15]. The normality of the data was checked by the Kolmogorov–Smirnov test. Since the test of normality did not show a normal distribution, continuous variables were presented as median (interquartile range) while categorical variables were presented as proportion (95% confidence interval). The Kruskal–Wallis test, Mann Whitney U test, Chi-squared test and two-population proportion test were applied to determine the statistical differences between groups. Associated cardiovascular risk factors were investigated using multivariable logistic regression. All analyses were executed using Statistical Package for the Social Sciences (SPSS version 20). A P-value $<0.05$ was deemed statistically significant.

## Results

Table 1 shows the comparison of characteristics of subjects who participated in the 2001–02, 2009–10 and 2016–17 surveys. Proportion of men/women in 2001–02, 2009–10 and 2016–17 were 32.3/67.7, 34.6/65.4 and 51.4/48.6 with median (IQR) age of 32(25–45), 40(30–50) and 48

**Table 1. General characteristics of studied sample.**

| Variables | Survey year | | | P-value |
|---|---|---|---|---|
| | **2001–2002** | **2009–2010** | **2016–2017** | |
| n | 2515 | 1377 | 358 | |
| **Gender** | | | | |
| Male | 32.3(30.5–34.2) | 34.6(32.2–37.2) | 51.4(46.2–56.5)[ab] | <0.0001 |
| Female | 67.7(65.8–69.5) | 65.4(62.8–67.8) | 48.6(43.5–53.8)[ab] | |
| **Age (years)** | 32(25–45) | 40(30–50)[a] | 48(38–57)[ab] | <0.0001 |
| 20–29 | 34(32.2–35.9) | 22.9(20.8–25.2)[a] | 6.1(4.1–9.2)[ab] | <0.0001 |
| 30–39 | 28.6(26.9–30.4) | 25.9(23.6–28.2) | 22.3(18.3–27)[a] | |
| 40–49 | 20.1(18.5–21.7) | 21.2(19.1–23.4) | 28.5(24.1–33.4)[ab] | |
| 50–59 | 11(9.8–12.3) | 18.4(16.5–20.6)[a] | 21.8(17.8–26.4)[a] | |
| ≥60 | 6.3(5.4–7.4) | 11.5(10–13.3)[a] | 21.2(17.3–25.8)[ab] | |
| **Marital status** | | | | |
| Single | - | 15.5(13.7–17.5) | 5.7(3.7–8.7)[b] | <0.0001 |
| Married | - | 83.9(81.8–85.7) | 93.2(90–95.4)[b] | |
| Divorced/separated/widow | - | 0.7(0.3–1.3) | 1.1(0.4–3) | |
| **Body mass index (kg/m2)** | 19.8(17.1–23.1) | 24.8(21–28.7)[a] | 25.4(23.2–28.6)[ab] | <0.0001 |
| <23 kg/m$^2$ | 72.9(71.1–74.6) | 37.5(35–40.1)[a] | 21.8(17.8–26.4)[ab] | <0.0001 |
| 23–24.9 kg/m$^2$ | 10.9(9.7–12.2) | 15.7(13.9–17.7)[a] | 24(19.9–28.7)[ab] | |
| ≥25 kg/m$^2$ | 16.2(14.8–17.7) | 46.8(44.2–49.5)[a] | 54.2(49–59.3)[ab] | |
| Waist circumference (cm) | - | 90(42–100) | 97(91–107)[b] | <0.0001 |
| Systolic blood pressure (mmHg) | 110(100–120) | 120(120–140)[a] | 130(120–140)[ab] | <0.0001 |
| Diastolic blood pressure (mmHg) | 70(60–80) | 80(80–90)[a] | 90(80–100)[b] | <0.0001 |
| **Education** | | | | |
| Illiterate/literate | - | 57(54.4–59.6) | 49.6(44.3–54.9)[b] | <0.0001 |
| Primary/secondary | - | 36.2(33.7–38.8) | 36.3(31.3–41.5) | |
| High school and above | - | 6.8(5.6–8.2) | 14.2(10.8–18.3)[b] | |
| **Exercise** | | | | |
| Sedentary to light | - | 87(85.2–88.7) | 81.2(72.2–87.9)[b] | 0.016 |
| Moderate to heavy | - | 13(11.3–14.8) | 18.8(12.1–27.8)[b] | |
| **Family history of diabetes** | | | | |
| No | 99.1(98.6–99.4) | 68.2(65.7–70.6)[a] | 54.2(49–59.3)[ab] | <0.0001 |
| Yes | 0.9(0.6–1.4) | 31.8(29.4–34.3)[a] | 45.8(40.7–51)[ab] | |
| Cholesterol (mg/dl) | - | 157(133–186.8) | 169.5(139.3–202.8)[ab] | <0.0001 |
| Triglyceride (mg/dl) | - | 145(104–211) | 158(107.3–250.5)[b] | 0.005 |
| HDL (mg/dl) | - | 36(30–49) | 29(20.4–38)[b] | <0.0001 |
| LDL (mg/dl) | - | 87(70–108) | 104(84–128)[b] | <0.0001 |
| Cholesterol/HDL ratio | - | 4.3(3.2–5.2) | 5.8(4.3–7.8)[b] | <0.0001 |

Data presented as proportion (95% confidence interval) or Median (interquartile range)

[a] Significantly different from Survey 2001–2002

[b] Significantly different from Survey 2009–2010

Mann Whitney U test and two population proportion tests were applied

(38–57) years respectively. Pronounced increases were seen in body mass index, waist circumference, blood pressure and family history of diabetes. A marked worsening in lipid profile was observed in 2016–2017 as compared to previous years (P-value<0.0001). In 2016–17, the proportion of illiteracy substantially decreased and the percentage of high school and above

education improved as compared to 2009–10. Additionally, it was observed that people were more physically active in 2016–17 as compared to 2009–10 (P-value<0.05).

Table 2 shows the age-standardised prevalence of each cardiovascular risk factor for each survey period. The prevalence of diabetes was 4.2 (3.4–4.9), 7.8 (6.6–9.2), and 31.9 (26.9–37.4) in the year 2001–02, 2009–10 and 2016–17 respectively. Notably, overall prevalence increased compared with previous years (P-value<0.0001). Among men, prevalence of diabetes in 2001–

**Table 2. Trends in prevalence of diabetes, obesity, hypertension, dyslipidaemia and tobacco and alcohol use.**

| Variables | Survey year | | | Percentage point change from 2001–2002 to 2016–2017, % (95% CI) |
|---|---|---|---|---|
| | 2001–2002 | 2009–2010 | 2016–2017 | |
| n | 2515 | 1377 | 358 | |
| **Diabetes** | | | | |
| Overall | 4.2(3.4–4.9) | 7.8(6.6–9.2)[a] | 31.9(26.9–37.4)[ab] | 27.7(24.7–30.6) |
| Male | 6.9(5.4–8.7) | 8.2(6.3–10.8) | 38.5(31–46.7)[ab] | 31.6(28.1–35.0) |
| Female | 2.9(2.2–3.8) | 7.5(6.1–9.3)[a] | 25.5(19.3–32.9)[ab] | 22.6(20.0–25.1) |
| Self-reported | 2.7(2.1–3.3) | 5.5(4.4–6.7)[a] | 19.8(15.8–24.7)[ab] | 17.1(14.7–19.4) |
| New | 1.5(1.1–2) | 2.3(1.7–3) | 12.1(8.9–16.1)[ab] | 10.6(8.7–12.4) |
| **Pre-diabetes** | | | | |
| Overall | 1.7(1.3–2.2) | 3.6(2.8–4.6)[a] | 10.7(7.6–14.9)[ab] | 9.0(7.1–10.8) |
| Male | 2.2(1.5–3.4) | 3(2–4.4) | 12.3(7.7–19)[ab] | 10.1(8.0–12.1) |
| Female | 1.4(1–2.1) | 3.9(2.8–5.3)[a] | 9.2(5.6–14.9)[ab] | 7.8 (6.1–9.4) |
| **Hypertension** | | | | |
| Overall | 3.7(3.1–4.5) | 25.2(22.9–27.6)[a] | 32.3(27.3–37.6)[ab] | 28.6(25.7–31.4) |
| Self-reported | 0(0–0) | 19.8(17.7–22) | 8(5.5–11.5)[b] | 8(5.5–11.5) |
| Newly diagnosed | 3.7(3.1–4.5) | 5.4(4.3–6.9)[a] | 24.3(20–29.2)[ab] | 12(7–18) |
| **Overweight/obese** | 26.5(24.7–28.3) | 61.7(58.8–64.4)[a] | 76.5(71–81.2)[ab] | 50(44.8–55.2) |
| **Central obesity** | - | 60.6(57.7–63.4) | 84.1(78.9–88.2)[b] | 23.5(17.9–29)* |
| **Combined obesity** | 15.7(14.3–17.2) | 76.2(73.6–78.5)[a] | 88.1(83.4–91.5)[ab] | 72.4(67.6–77.1) |
| **Cholesterol≥200 mg/dl** | - | 15.4(13.2–17.8) | 27.1(22.1–32.7)[b] | 11.7(7.2–16.1)* |
| **TC/HDL ratio ≥5.9** | - | 12.5(10.6–14.7) | 47.2(41.3–53.3)[b] | 34.7(30–39.3)* |
| **Dyslipidaemia** | - | 82.2(79.6–84.6) | 93.5(89.5–96)[b] | 11.3(7.1–15.5)* |
| **Tobacco addiction** | | | | |
| Ex-addict | 0(0–0) | 3.2(2.4–4.2) | 4.2(2.7–6.6) | 4.2(2.7–6.6) |
| Current addict | 1.5(1–2.2) | 43.4(40.6–46.3)[a] | 24.3(19.6–29.7)[ab] | 9.3(5.2–13.3) |
| **Alcohol addiction** | | | | |
| Current addict | - | 7.7(6.3–9.4) | 1.1(0.4–2.9)[b] | 6.6(3.7–9.4)* |
| Ex-addict | - | 1.9(1.3–2.6) | 5.7(3.7–8.7)[b] | 3.8(1.9–5.6)* |

Data presented as age-adjusted prevalence % (95% confidence interval)

[a] Significantly different from Survey 2001–2002

[b] Significantly different from Survey 2009–2010

*Percentage point difference from 2009–2010 to 2016–2017

Two population proportion test was applied

Diabetes was defined as FPG≥126mg/dl; Pre-diabetes defined as FPG:110–125 mg/dl

New hypertension was defined as a systolic blood pressure ≥140mm Hg and/or diastolic blood pressure ≥90mm Hg; known hypertensive was defined as if already diagnosed by a physician or if taking any antihypertensive medications.

Overweight/obese was defined as BMI≥23 kg/m$^2$; central obesity was defined as waist circumference ≥90cm and ≥80cm in males and females, respectively; combined obesity was defined as BMI≥25 kg/m$^2$ or waist circumference ≥90cm and ≥80cm in males and females, respectively.

Dyslipidaemia was defined as serum cholesterol >200 mg/dL or serum LDL-C >130 mg/dL or serum HDL-C <40 mg/dL and <50 mg/dL for male and female respectively, or serum triglycerides >150 mg/dL.

02 and 2009–10 was similar, and significantly lower than 2016–17, whereas among women the prevalence of diabetes was consistently increased with time (P-value<0.05). Similarly, self-reported diabetes increased over time, whilst newly diagnosed diabetes was only higher in 2016–17 as compared to previous years.

A similar pattern was observed for pre-diabetes. Overall prevalence of pre-diabetes was observed 1.7 (1.3–2.2), 3.6 (2.8–4.6) and 10.7 (7.6–14.9) in the year 2001–02, 2009–10 and 2016–17 respectively. Throughout the observed time period, a rapid increase was also observed in hypertension, obesity and dyslipidaemia. However, the prevalence of current addiction to tobacco and alcohol decreased.

Fig 1 shows the increasing trends of diabetes during the study period. Among the age group 20–39 years, prevalence of diabetes was stable in 2001–02 until 2009–10 then gradually rose with increasing age and time, however in year 2016–17 a significant increase in the prevalence of diabetes was seen from very early ages (P-value<0.0001). Similarly, the prevalence of pre-diabetes showed a remarkable rise in all age groups with respect to the whole study period except for age group 40–49 years during 2001–02 and 2009–10.

Prevalence of hypertension was especially lower in 2001–02 as compared to 2009–10 and 2016–17 among all age groups (P-value<0.0001). No significant differences were observed in the prevalence of hypertension between 2009–10 and 2016–17 in all age groups except for age group 30–39 years (P-value<0.05). Moreover, the prevalence of subjects defined as being over-weight/obese, having central obesity, and those with dyslipidaemia showed an increasing trend in all age groups (P-value<0.0001).

Notably, the prevalence of tobacco addiction increased from 2001–02 until 2009–10; it declined among all age groups in 2016–17. Prevalence of alcohol addiction was also notably reduced in 2016–17 as compared to 2009–10 in age group 20–39 and ≥60 years. No trend in alcohol addiction was identified in age group 40–59 years.

Fig 2 shows pooled prevalence of pre-diabetes and diabetes with respect to gender. At age group 40–59 years, the prevalence of pre-diabetes was slightly higher among men as compared to women. The prevalence of diabetes started increasing among men at the early age of 30, and increased with increasing age.

Table 3 shows the multivariable logistic regression for identifying the associated risk factors for diabetes, pre-diabetes, hypertension, obesity, central obesity, and dyslipidaemia.

Pre-diabetes was associated with increasing age and hypertension. Dyslipidaemia was associated with male gender, central obesity and hypertension. Being overweight/obese was associated with being married, high school education, positive family history of diabetes, central obesity and hypertension. Diabetes was associated with age, male gender, being married, high school education, positive family history of diabetes, no addiction to tobacco and alcohol, central obesity and hypertension. Central obesity was associated with dyslipidaemia, diabetes, being overweight/obese and no addiction to tobacco and alcohol, whereas hypertension was associated with age, female gender, positive family history of diabetes, current or ex-addiction of tobacco and alcohol and being overweight/obese. All in all, a complex interlink was observed between all CVD risk factors (P-value<0.05).

## Discussion

The overall prevalence of diabetes and prediabetes in Baluchistan has increased precipitously in 2016–17 as compared with the previous survey years in 2009–10 and 2001–02. Among the age groups, prevalence of diabetes was drastically increased, especially at the age of 30–39 years in 2016–17. Similarly, prevalence of pre-diabetes was a rising trend in all age groups with respect to the study period, however the 20–30 years age group had a higher prevalence of

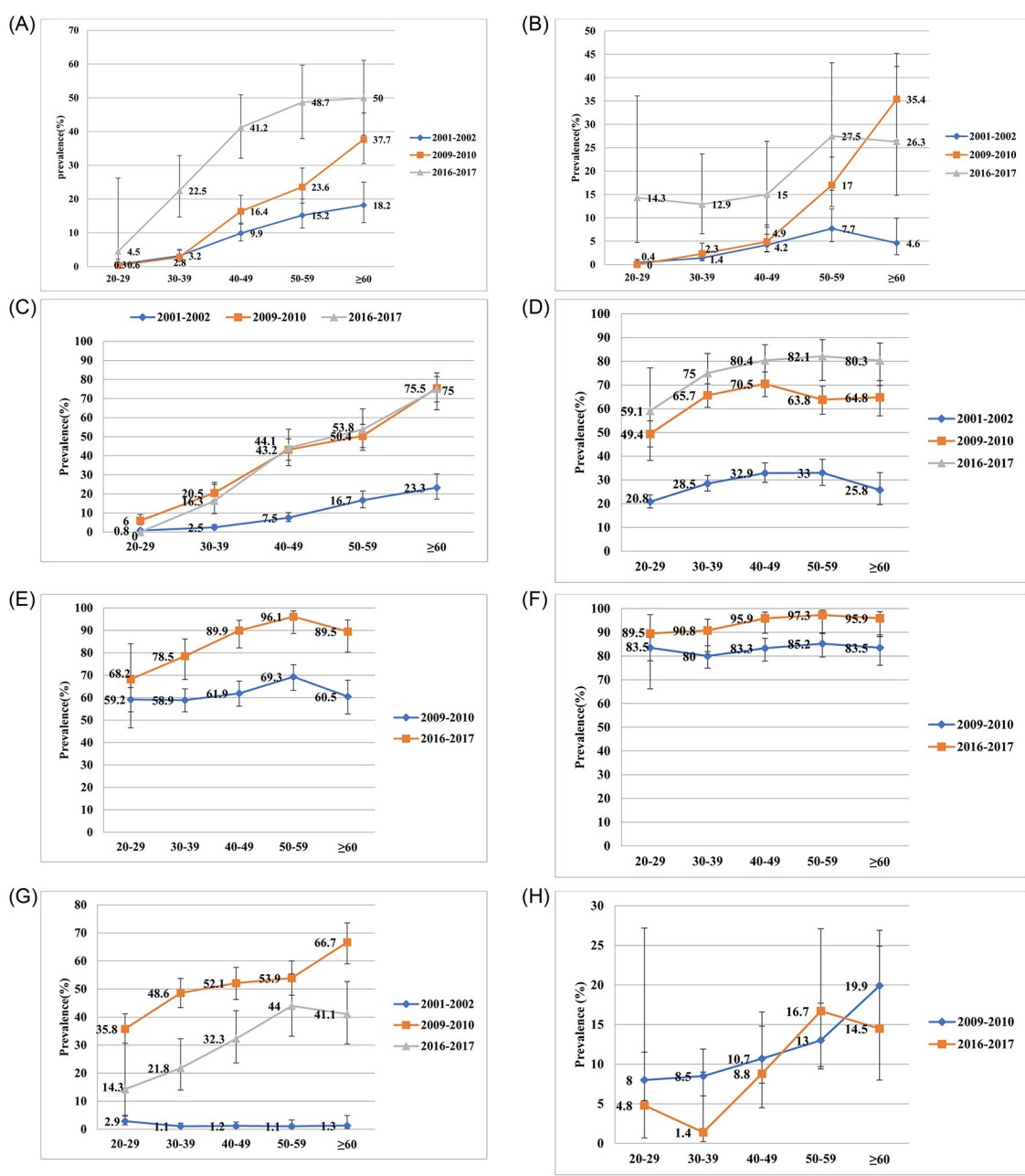

**Fig 1. A:** Age-specific prevalence of Diabetes. Diabetes was defined as FPG≥126mg/dl. **B:** Age-specific prevalence of Pre-diabetes. Pre-diabetes defined as FPG:110–125 mg/dl. **C:** Age-specific prevalence of hypertension. Hypertension was defined as if the systolic blood pressure was ≥140mm Hg and/or diastolic blood pressure ≥90mm Hg or, already diagnosed by a physician or taking any antihypertensive medication. **D:** Age-specific prevalence of Overweight/obese. Overweight/obesity as defined as BMI≥23 kg/m². **E:** Age-specific prevalence of Central obesity. Central obesity was defined as waist circumference ≥90cm and ≥80cm in males and females. **F:** Age-specific prevalence of Dyslipidaemia. Dyslipidaemia was defined as cholesterol>200mg/dl or triglyceride>150 mg/dl or LDL>. **G:** Age-specific prevalence of tobacco addiction. **H:** Age-specific prevalence of alcohol addiction.

prediabetes in 2016–17. Taken together, a complex interlink was observed between all CVD risk factors with diabetes.

The trends of diabetes and prediabetes in 2016–17 rose by more than two-fold in glycaemic dysregulation compared to 2001–10. Various other surveys have shown higher prevalence of

(A)

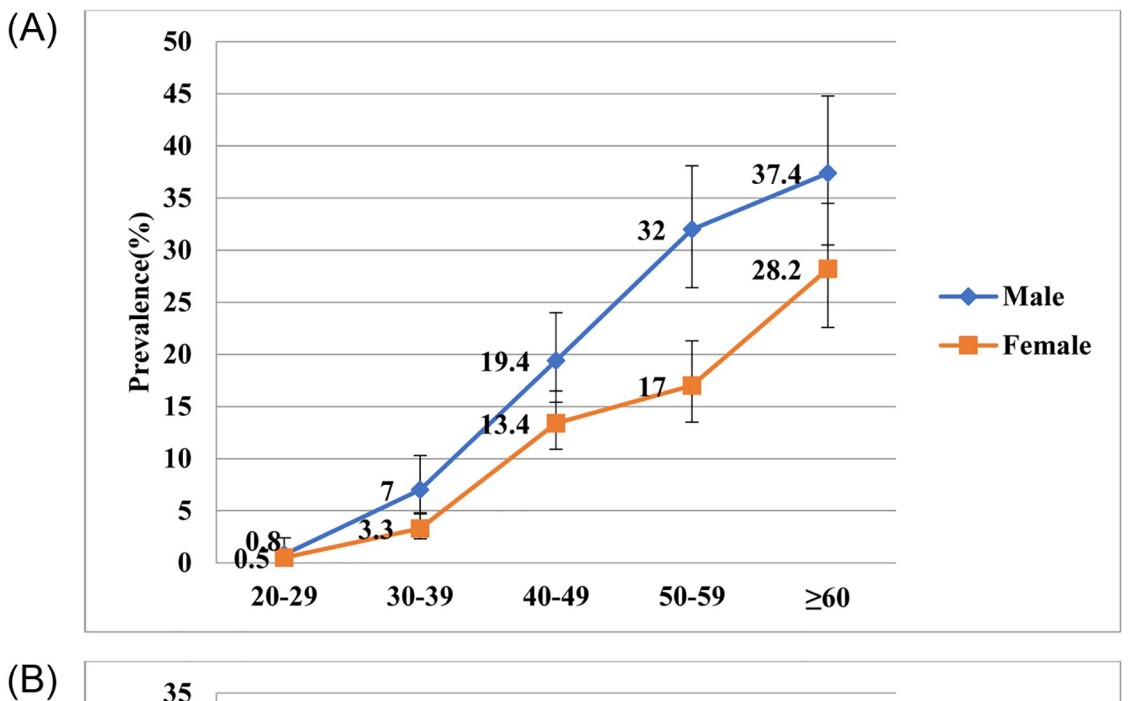

(B)

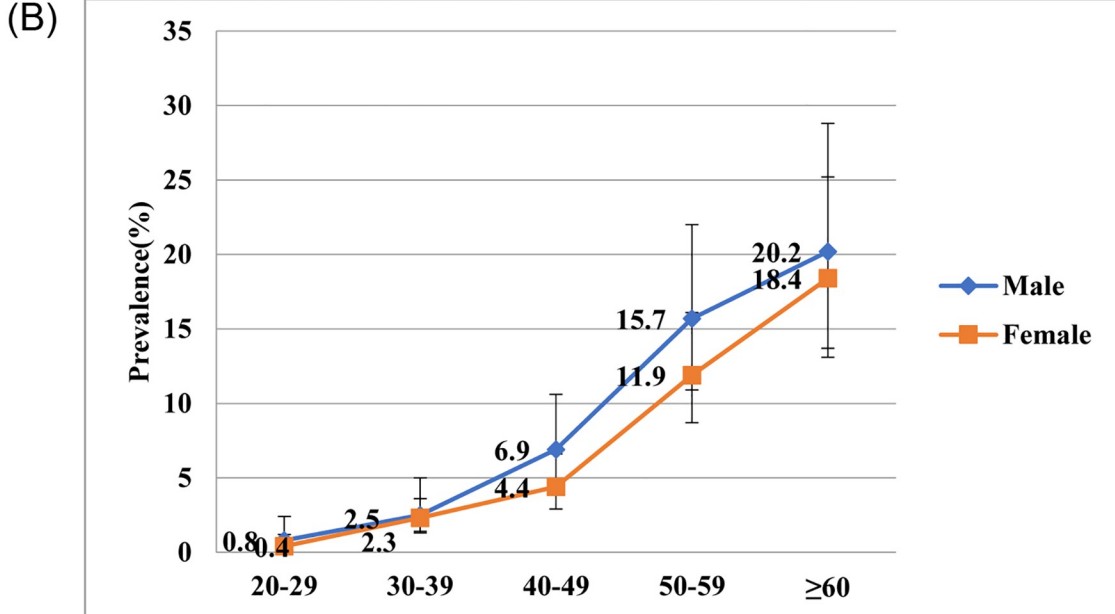

**Fig 2. A:** Age specific prevalence of diabetes from year 2001–2017. **B:** Age specific prevalence of pre-diabetes from year 2001–2017.

diabetes in recent decades which support our findings [24, 25]. A possible explanation is the rapid transition to diabetes from pre-diabetes; for example, in the younger age groups, early onset of diabetes occurs which leads to diabetes later on in the age group 30 to 40 years. Previously, it was noted that in the province of Baluchistan the diabetes to pre-diabetes ratio is almost 1:2, suggesting that a large number of individuals are at risk of developing T2DM [24, 26]. Overall, the prevalence of diabetes and prediabetes was much higher among males in 2001–17. In the age group 40–59 years, the prevalence of pre-diabetes was marginally higher among men as compared to women. The prevalence of diabetes started increasing among men

**Table 3. Adjusted odd ratios of potential risk factors associated with diabetes.**

| Factors | Pre-diabetes OR (95% CI) | P-value | Diabetes OR (95% CI) | P-value | Hypertension OR (95% CI) | P-value | Overweight/Obesity OR (95% CI) | P-value | Central obesity OR (95% CI) | P-value | Dyslipidaemia OR(95% CI) | P-value |
|---|---|---|---|---|---|---|---|---|---|---|---|---|
| Age | 1.08(1.05–1.1) | <**0.0001** | 1.06(1.04–1.08) | <**0.0001** | 1.08(1.07–1.1) | <**0.0001** | 0.99(0.98–1.01) | 0.26 | 1(0.99–1.01) | 0.848 | 1.01(0.99–1.02) | 0.476 |
| Male | 1.2(0.56–2.56) | 0.638 | 2.5(1.6–3.89) | <**0.0001** | 0.56(0.38–0.83) | **0.004** | 1(0.71–1.41) | 0.995 | 1.19(0.84–1.69) | 0.322 | 1.65(1.01–2.71) | **0.046** |
| Married | 2.23(0.28–17.55) | 0.445 | 5.16(1.19–22.4) | **0.028** | 0.93(0.52–1.67) | 0.806 | 1.58(1.05–2.4) | **0.03** | 1.02(0.66–1.58) | 0.915 | 0.77(0.43–1.39) | 0.386 |
| Divorced/ separated /widowed | 3.49(0.15–79.12) | 0.433 | 5.93(0.63–55.82) | 0.12 | 1.57(0.31–8.05) | 0.587 | 0.99(0.26–3.83) | 0.987 | 1.59(0.38–6.72) | 0.527 | 0(0–0) | 0.999 |
| Primary/ secondary education | 0.67(0.34–1.3) | 0.234 | 1.12(0.74–1.69) | 0.597 | 0.99(0.72–1.37) | 0.956 | 1.14(0.86–1.52) | 0.359 | 0.9(0.67–1.2) | 0.461 | 1.27(0.86–1.87) | 0.237 |
| High school or above | 2.8(0.94–8.31) | 0.064 | 2.31(1.15–4.63) | **0.018** | 0.76(0.39–1.46) | 0.402 | 2.95(1.44–6.04) | **0.003** | 1.76(0.91–3.41) | 0.094 | 0.88(0.36–2.12) | 0.768 |
| Exercise moderate to high | 0.86(0.32–2.27) | 0.754 | 0.7(0.38–1.29) | 0.256 | 1.02(0.62–1.66) | 0.947 | 0.99(0.65–1.52) | 0.977 | 0.72(0.47–1.1) | 0.13 | 1.71(0.91–3.23) | 0.096 |
| Positive family history of DM | 1.01(0.58–1.79) | 0.963 | 1.73(1.2–2.5) | **0.004** | 2.09(1.55–2.82) | <**0.0001** | 1.4(1.06–1.84) | **0.019** | 0.8(0.61–1.06) | 0.116 | 0.92(0.64–1.33) | 0.649 |
| Ex or current tobacco addicted | 0.69(0.39–1.25) | 0.22 | 0.68(0.46–0.99) | **0.044** | 1.48(1.08–2.01) | **0.014** | 1.16(0.88–1.53) | 0.279 | 0.63(0.47–0.83) | **0.001** | 1(0.69–1.46) | 0.999 |
| Ex or current alcohol addicted | 1.15(0.46–2.87) | 0.765 | 0.4(0.21–0.78) | **0.006** | 1.63(0.97–2.73) | 0.066 | 1.02(0.64–1.62) | 0.928 | 0.53(0.33–0.83) | **0.006** | 0.43(0.23–0.79) | **0.007** |
| Overweight or obese | 1.34(0.75–2.41) | 0.322 | 1.21(0.81–1.79) | 0.351 | 1.78(1.31–2.44) | <**0.0001** | - | - | 1.55(1.18–2.02) | **0.002** | 1.39(0.98–1.97) | 0.062 |
| Central obesity | 0.91(0.52–1.6) | 0.739 | 1.61(1.08–2.38) | **0.018** | 0.83(0.61–1.14) | 0.245 | 1.53(1.17–2.01) | **0.002** | - | - | 6.37(4.33–9.38) | <**0.0001** |
| Hypertension | 2.15(1.17–3.95) | **0.014** | 2.37(1.6–3.52) | <**0.0001** | - | - | 1.7(1.25–2.3) | **0.001** | 0.84(0.62–1.14) | 0.259 | 1.81(1.19–2.75) | **0.006** |
| Dyslipidaemia | 2.4(0.92–6.27) | 0.074 | 0.76(0.45–1.28) | 0.301 | 1.77(1.15–2.72) | **0.009** | 1.36(0.96–1.93) | 0.085 | 6.25(4.25–9.19) | <**0.0001** | - | - |
| Diabetes | - | - | - | - | 2.37(1.61–3.49) | <**0.0001** | 1.17(0.79–1.72) | 0.43 | 1.75(1.17–2.6) | **0.006** | 0.76(0.45–1.28) | 0.298 |

Marital status: single; Education: Illiterate/can read/write; Exercise: Sedentary to light; Tobacco: No addiction; Alcohol: No addiction,

Obesity: Normal; Central obesity: Normal; Hypertension: No; Dyslipidaemia> No; Diabetes: No

AOR: adjusted odd ratios; CI: confidence interval;

Factors which were associated in univariate analysis were entered in the final model:

P-value<0.05 considered to be statistically significant

at the early age of 30, and it increased with increasing age. The high prevalence of T2DM in men is similar to some other regions of the world, where numbers are continuing to rise [25]. In men, it is mainly due to biological sex differences, and partially due to stress and related behavioural risk factors like smoking, alcohol consumption, or physical inactivity [27].

Previous studies have urged awareness with regards to the higher prevalence of risk factors leading to diabetes in Baluchistan in the last 20 years [15]. We also found a complex interlink with central obesity, obesity, dyslipidaemia and hypertension with diabetes, though the associated risk factors showed the increasing trend in all age groups. However, the age group 40 to 50 years was highly associated with CVD risk factors and considered as a high-risk age group. Most of the associated risk factors are modifiable. Our findings validate the necessity of risk assessment, early screening, surveillance programs and appropriate lifestyle interventions which improve the health profile of the population by improving modifiable risk factors in this age group [28].

We also found a dramatically rising trend of obesity in the 20 to 50 years age group in all survey years, which could be considered the cause of a rising prevalence of early onset of

diabetes. It is a known fact that dyslipidaemia often increases the chances of obesity and obesity increases the risk of dyslipidaemia [29]. In our study, central obesity is highly associated with dyslipidaemia and diabetes. It indicates that central obesity is far more sensitive for this population. The role of obesity was also more pronounced for women than men, reflecting the high prevalence in women as is consistent with the Kanter et al. study [30].

There was a stark difference in prevalence of hypertension in the age group 30–39 years, highlighting the escalating burden of the "silent killer" in this age group. However, in the 2016–17 survey, new hypertensives were reported as more prevalent as compared to previous surveys. This is mostly likely due to social and cultural differences, dietary and lifestyle factors and lack of awareness and treatment of hypertension in Baluchistan. It was also noted that with increasing age, trends of hypertension also increase. This difference may be due to stiffness in the aorta and arterial walls that occurs with increasing age, contributing to rising hypertension in older age [31]. It may additionally be due to inflammation and oxidative stress that occurs with ageing [32]. We noted hypertension was associated with age, female gender, positive family history of diabetes, current or ex-addiction of tobacco and alcohol and being overweight/obese, similar to the Singh et al. study [33].

Moreover, differences in the dyslipidaemia trend were seen in the 30 years age group and rising with increasing age from 2010–17. This rising dyslipidaemia trend in the young age group of 30 years was similar to a recently reported study from China [34]. The trend increased with increasing age and is considered a major risk factor for atherosclerotic cardiovascular disease (ASCVD) in older age groups [35]. Dyslipidaemia needs early management in diabetics, especially in males under the age of 60 years and in post-menopausal females.

Smoking, alcohol addiction and physical inactivity have a small contribution as compared to other associated risk factors in T2DM incidence [23]. Throughout the observed time period, the prevalence of current addiction to tobacco and alcohol were significantly decreased showing a positive change in our study. However, our findings are in contrast to Saqib et al who showed the high prevalence of tobacco users in the general population [36]. It may be due to the fact that people with diabetes and overt complications stop smoking and alcohol intake. However, a rising trend in alcohol addiction was seen in age group 50–60 years. This could be due to easy accessibility and availability of tobacco and alcohol in Pakistan. Improved awareness is still required to stop advertisements, supply and events sponsored by tobacco and alcohol companies at a government level.

We also found a high prevalence of diabetes in people with a strong family history of diabetes indicating genetic predisposition [37]. So, to prevent diabetes, it is necessary to further research genetic and environmental influences. In Pakistan, diabetes and its complications have become a challenge for public health. Major contributions to the alarming rise in diabetes are adaptation to urbanization, unhealthy lifestyles, maternal and foetal malnutrition, and genetic or epigenetic factors, with additional risk from sedentary behaviour, poor nutrition, and excessive consumption of calories, salt, saturated fat, and sugar. According to a 1996 census, the Pakistani population was 127 million with an average annual growth rate of 4.8% [38, 39]. The most alarming fact is that 31% of the population lived below the poverty line, and 40% had limited to no access to even essential health services [40]. The government offered funding for diabetes mellitus (DM) as part of the overall healthcare budget, but just a small fraction is allocated. It is time to take appropriate actions to improve the existing infrastructure of healthcare services in order to reduce deaths and the disability burden as a result of diabetes. Policy-level actions and facilitations are necessary in reducing the lifestyle risk factors related to T2DM and to prevent premature morbidity and mortality. This includes reducing consumption of unhealthy food that is easily accessible and cheap, possibly by increasing taxation on sugar-sweetened beverages and introducing subsidies for healthier foods. Junk food, salty

snacks, sweets, deserts, cola consumption, alcoholic beverages, physical inactivity, along with genetic predisposition mainly leads to the onset of diabetes. However, in Baluchistan, the feasibility and political palatability of such approaches require further investigation. Furthermore, in Baluchistan, it is more important to implement primary prevention at all levels such as in schools, hospitals and workplaces. To prevent transgenerational obesity and diabetes, maternal and child health awareness and screening camps should be a top priority.

The limitations of this study include the small sample size and a wide variation in the sampling frame in survey year 2016–17 due to the selection of only subjects located in the rural area of Hub in the second National Diabetes Survey of Pakistan. Additionally, data regarding marital status, exercise, education and lipid profile were not present in the 2001–02 survey year and thus not compared. However, comparing three survey years to show trends in diabetes trend is a strength.

## Conclusion

Overall, rural Baluchistan-Pakistan is confronted with increasing trends of early onset of T2DM in recent decades due to highly associated CVD risk factors, especially central obesity and dyslipidaemia, and it raises a major public health challenge. Our findings suggest Pakistan urgently needs national strategies for the early diagnosis of diabetes, and large-scale implementation of cost-effective preventive, therapeutic interventions and management targeting T2DM and associated risk factors.

## Acknowledgments

We acknowledge the support of the following doctors of Balochistan surveys for their help in recruiting and conducting the studies: Prof. Rubina Hakeem, Dr. Muhammad Zafar Iqbal Hydrie, Dr. Khursheed Ahmed, Dr. Syed Faraz Danish Alvi, Prof. Muhammad Yakoob Ahmedani, and second National Diabetes Survey of Pakistan Members.

## Author Contributions

**Conceptualization:** Khalid Abdul Basit, Asher Fawwad, Bilal Tahir, Abdul Basit.

**Formal analysis:** Khalid Abdul Basit, Nida Mustafa, Bilal Tahir.

**Investigation:** Khalid Abdul Basit.

**Methodology:** Khalid Abdul Basit.

**Resources:** Abdul Basit.

**Software:** Nida Mustafa.

**Supervision:** Abdul Basit.

**Validation:** Khalid Abdul Basit, Nida Mustafa.

**Visualization:** Khalid Abdul Basit, Nida Mustafa.

**Writing – original draft:** Khalid Abdul Basit, Nida Mustafa.

**Writing – review & editing:** Asher Fawwad, Thomas Davey, Bilal Tahir, Abdul Basit.

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
