## [Decision Letter · Decision Letter 0]

23 Jan 2023

PONE-D-22-22497Changes in the prevalence of diabetes, prediabetes and associated risk factors in rural Baluchistan; a secondary analysis from repeated surveys (2002-2017)PLOS ONE

Dear Prof. Abdul Basit,

Thank you for submitting your manuscript to PLOS ONE. After careful consideration, we feel that it has merit but does not fully meet PLOS ONE’s publication criteria as it currently stands. Therefore, we invite you to submit a revised version of the manuscript that addresses the points raised during the review process.

Dear authors,

I commend that the topic of the paper is interesting, especially for the alarming increase in a rural population. However, my own reading raised a number of concerns which need to be addressed before any final decision is made. First, the Methods section needs clarification. For instance, it is not clear which outcomes were collected in three surveys and which ones were not. Instances are shown in Figure 1E and 1F where only two surveys are presented. Second, authors refer to "pooled prevalence", it seems that they just collapsed pre and diabetes categories? Third, the methods section is fuzzy the way it stands now. It is unclear which are the outcomes and associated factors in the paper. This section needs to reworked to reflect the standard of PLOS One. Third, this section is unclear regarding the sampling design in the three surveys, and how the sampling design was taken into account the estimations. Without this information or clarification, it is not possible for the readers to trust the estimates. Fourth, I wonder how readers could trust estimates from the 2001-2002 survey given tremendous differences with 2009-2010 and 2016-2017 surveys. The estimates from these surveys cannot be trusted without further explanations of these differences. What happened between 2001-2002 survey and subsequent surveys. I question what has changed in the Pakistani rural context to see such a shift in only 10 years. Usually, such changes are expected on long-run but short- or mid-run. As a consequence, most differences in percentage points between 2001-2002 and 2016-2017 are above 10 points and combined obesity, over 15 years, increased by 72 percentage points. This is impressive and questionable at same time (Table 2). Surprisingly, the Discussion section is silent about convincing explanation of this health catastrophe. This is where authors need to be more creative and build from their knowledge of the ground in the Pakistani context to explain to the readers what happened to explain these findings. Finally, my fears are consolidated by Table 3 where only a few variables showed statistical significance with the outcomes of the paper. Why?

We look forward to receiving your revised manuscript.

Kind regards,

Zacharie Tsala Dimbuene, Ph.D.

Academic Editor

PLOS ONE

2. PLOS requires an ORCID iD for the corresponding author in Editorial Manager on papers submitted after December 6th, 2016. Please ensure that you have an ORCID iD and that it is validated in Editorial Manager. To do this, go to ‘Update my Information’ (in the upper left-hand corner of the main menu), and click on the Fetch/Validate link next to the ORCID field. This will take you to the ORCID site and allow you to create a new iD or authenticate a pre-existing iD in Editorial Manager. Please see the following video for instructions on linking an ORCID iD to your Editorial Manager account: https://www.youtube.com/watch?v=_xcclfuvtxQ.

3. Please ensure that you refer to Figure1 & 2 in your text as, if accepted, production will need this reference to link the reader to the figure.

Reviewers' comments:

Reviewer's Responses to Questions

**Comments to the Author**

1. Is the manuscript technically sound, and do the data support the conclusions?

Reviewer #1: Yes

Reviewer #2: Yes

2. Has the statistical analysis been performed appropriately and rigorously? 

Reviewer #1: Yes

Reviewer #2: Yes

3. Have the authors made all data underlying the findings in their manuscript fully available?

Reviewer #1: Yes

Reviewer #2: Yes

4. Is the manuscript presented in an intelligible fashion and written in standard English?

Reviewer #1: Yes

Reviewer #2: Yes

5. Review Comments to the Author

Reviewer #1: Diabetes is one of the biggest challenges for the public health and the number of people affected by this diseases still increases. There is still a big gap in diabetes management between high and low-income countries. It has been reported that approximately 80% of adults with diabetes worldwide live in low-income and middle-income countries.

Therefore, despite the fact that presented study comes from one region with small study group, I think it is valuable to present. Most pre-existing studies come from urban populations or high income countries, so in that way presented study adds to the existing evidence and helps to draw attention to the problem of diabetes in rural areas. In my opinion the article is well-written. I have some minor comments:

I think it would be helpful if overall number of participants was mentioned in the abstract and the methods section.

It is stated that “BMI <23 kg/m2 was considered as normal, ≥23-24.9 kg/m2 as overweight and ≥25 kg/m2 as obese”. The citation [19] at the end of the sentence doesn’t lead to the source of the criteria and they differ from the widely recognizable WHO norms. The reference should indicate the guidelines for BMI, if the Authors chose the norms for Asia-Pacific region, it should be clearly explained.

The abstract should be improved, as there are some nominal sentences.

Reviewer #2: In page 3, line 65, It is mentioned that population of Baluchistan (85% rural, 38% urban) which needs correction

In discussion, page 18, findings of tobacco consumption at national level like in NCDs survey [https://pubmed.ncbi.nlm.nih.gov/29658619/] and Global Adult Tobacco Survey [https://pubmed.ncbi.nlm.nih.gov/29059338/] may be discussed in detail that why it is different from general population.

6. PLOS authors have the option to publish the peer review history of their article (what does this mean?). If published, this will include your full peer review and any attached files.

Reviewer #1: No

Reviewer #2: **Yes: **Dr. Ibrar Rafique

---

## [Author Response · Author response to Decision Letter 0]

10 Feb 2023

First, the Methods section needs clarification. For instance, it is not clear which outcomes were collected in three surveys and which ones were not. 

Response: Changes suggested are made in the manuscript and highlighted. 

Instances are shown in Figure 1E and 1F where only two surveys are presented. 

Response: Given the variables of central obesity and dyslipidemia were not obtained in survey year 2001-02 through the suggested changes suggested are being made in the methodology and highlighted. 

Second, authors refer to "pooled prevalence", it seems that they just collapsed pre and diabetes categories? 

Response: Yes, it gives collapsed prevalence of pre and diabetes category. However, we decided to keep the figures 2A and 2B , to understand the prevalence by gender and age. The term pooled is removed from the figure as it is not suited. Changes suggested are being made in the manuscript and highlighted.

Third, the methods section is fuzzy the way it stands now. It is unclear which are the outcomes and associated factors in the paper. This section needs to reworked to reflect the standard of PLOS One. Third, this section is unclear regarding the sampling design in the three surveys, and how the sampling design was taken into account the estimations. Without this information or clarification, it is not possible for the readers to trust the estimates. 

Response: Changes suggested are being made in the manuscript and highlighted.

Fourth, I wonder how readers could trust estimates from the 2001-2002 survey given tremendous differences with 2009-2010 and 2016-2017 surveys. The estimates from these surveys cannot be trusted without further explanations of these differences. What happened between 2001-2002 survey and subsequent surveys. I question what has changed in the Pakistani rural context to see such a shift in only 10 years. Usually, such changes are expected on long-run but short- or mid-run. As a consequence, most differences in percentage points between 2001-2002 and 2016-2017 are above 10 points and combined obesity, over 15 years, increased by 72 percentage points. This is impressive and questionable at same time (Table 2). Surprisingly, the Discussion section is silent about convincing explanation of this health catastrophe. This is where authors need to be more creative and build from their knowledge of the ground in the Pakistani context to explain to the readers what happened to explain these findings. 

Response: Changes suggested to improve the discussion are being made in the manuscript and highlighted.

Finally, my fears are consolidated by Table 3 where only a few variables showed statistical significance with the outcomes of the paper. Why?

Response: 

Though the odds prediabetes and diabetes had weak significance with covariates, but the cardiometabolic risk factors were strongly associated with hypertension, central obesity and dyslipidemia which implies policy decisions to tackle modifiable risk factors.

Reviewer #1: Diabetes is one of the biggest challenges for the public health and the number of people affected by this disease still increases. There is still a big gap in diabetes management between high and low-income countries. It has been reported that approximately 80% of adults with diabetes worldwide live in low-income and middle-income countries.

Therefore, despite the fact that presented study comes from one region with small study group, I think it is valuable to present. Most pre-existing studies come from urban populations or high income countries, so in that way presented study adds to the existing evidence and helps to draw attention to the problem of diabetes in rural areas. In my opinion the article is well-written. I have some minor comments:

I think it would be helpful if overall number of participants was mentioned in the abstract and the methods section.

Response: Changes suggested are made in the manuscript and highlighted.

It is stated that “BMI <23 kg/m2 was considered as normal, ≥23-24.9 kg/m2 as overweight and ≥25 kg/m2 as obese”. The citation [19] at the end of the sentence doesn’t lead to the source of the criteria and they differ from the widely recognizable WHO norms. The reference should indicate the guidelines for BMI, if the Authors chose the norms for Asia-Pacific region, it should be clearly explained.

Response: Changes suggested are made in the manuscript and highlighted.

The abstract should be improved, as there are some nominal sentences.

Response: Changes suggested are made in the manuscript and highlighted.

Reviewer #2: In page 3, line 65, It is mentioned that population of Baluchistan (85% rural, 38% urban) which needs correction

Response: Changes suggested are made in the manuscript and highlighted.

In discussion, page 18, findings of tobacco consumption at national level like in NCDs survey [https://pubmed.ncbi.nlm.nih.gov/29658619/] and Global Adult Tobacco Survey [https://pubmed.ncbi.nlm.nih.gov/29059338/] may be discussed in detail that why it is different from general population.

Response: Changes suggested are made in the manuscript and highlighted.

---

## [Decision Letter · Decision Letter 1]

3 Apr 2023

Changes in the prevalence of diabetes, prediabetes and associated risk factors in rural Baluchistan; a secondary analysis from repeated surveys (2002-2017)

PONE-D-22-22497R1

Dear Dr. Basit,

We’re pleased to inform you that your manuscript has been judged scientifically suitable for publication and will be formally accepted for publication once it meets all outstanding technical requirements.

Kind regards,

Zacharie Tsala Dimbuene, Ph.D.

Academic Editor

PLOS ONE

Additional Editor Comments (optional):

Reviewers' comments:

Reviewer's Responses to Questions

**Comments to the Author**

1. If the authors have adequately addressed your comments raised in a previous round of review and you feel that this manuscript is now acceptable for publication, you may indicate that here to bypass the “Comments to the Author” section, enter your conflict of interest statement in the “Confidential to Editor” section, and submit your "Accept" recommendation.

Reviewer #1: All comments have been addressed

Reviewer #2: (No Response)

2. Is the manuscript technically sound, and do the data support the conclusions?

Reviewer #1: Yes

Reviewer #2: Yes

3. Has the statistical analysis been performed appropriately and rigorously? 

Reviewer #1: Yes

Reviewer #2: Yes

4. Have the authors made all data underlying the findings in their manuscript fully available?

Reviewer #1: (No Response)

Reviewer #2: Yes

5. Is the manuscript presented in an intelligible fashion and written in standard English?

Reviewer #1: (No Response)

Reviewer #2: Yes

6. Review Comments to the Author

Reviewer #1: (No Response)

Reviewer #2: (No Response)

7. PLOS authors have the option to publish the peer review history of their article (what does this mean?). If published, this will include your full peer review and any attached files.

Reviewer #1: No

Reviewer #2: **Yes: **Dr. Ibrar Rafique

---

## [Editor Report · Acceptance letter]

11 Apr 2023

PONE-D-22-22497R1 

Changes in the prevalence of diabetes, prediabetes and associated risk factors in rural Baluchistan; a secondary analysis from repeated surveys (2002-2017) 

Dear Dr. Basit:

I'm pleased to inform you that your manuscript has been deemed suitable for publication in PLOS ONE. Congratulations! Your manuscript is now with our production department. 

Kind regards, 

on behalf of

Prof. Zacharie Tsala Dimbuene 

Academic Editor

PLOS ONE